# Low-Dose Propranolol as Secondary Prophylaxis for Varix Bleeding Decreases Mortality and Rebleeding Rate in Patients with Tense Ascites

**DOI:** 10.3390/jcm8050573

**Published:** 2019-04-26

**Authors:** Jin Hwa Park, Dae Won Jun, Jun Choi, Dong Hee Koh, Jai Hoon Yoon, Kang Nyeong Lee, Hang Lak Lee, Oh Young Lee, Byung Chul Yoon, Ho Soon Choi

**Affiliations:** 1Department of Internal Medicine, Hanyang University School of Medicine, Seoul 04763, Korea; pjh6718@hanmail.net (J.H.P.); jaihoonyoon@hanyang.ac.kr (J.H.Y.); leekn@hanyang.ac.kr (K.N.L.); alwayshang@hanyang.ac.kr (H.L.L.); leeoy@hanyang.ac.kr (O.Y.L.); yoonbc@hanyang.ac.kr (B.C.Y.); hschoi96@hanyang.ac.kr (H.S.C.); 2Department of Fusion Data Analytics, School of Industrial Management Engineering, Korea University, Seoul 02841, Korea; jun.choi@outlook.com; 3Department of Internal Medicine, Hallym University Dongtan Sacred Heart Hospital, Hwaseong-si 14103, Korea; dhkoh@hallym.or.kr

**Keywords:** propranolol, ascites, mortality

## Abstract

Background and Aim: The risk and benefit of non-selective propranolol in patients with tense ascites are controversial. This study aimed to investigate the effect of propranolol as secondary prophylaxis on varix rebleeding and overall mortality in patients with tense ascites. Methods: This study used a database of the Health Insurance Review and Assessment Service (HIRAS), which provides health insurance to 97.2% of the total population in Korea. A total of 80,071 patients first variceal bleeding as the first decompensated complication enrolled from 2007 to 2014. Results: There were 2274 patients with large-volume ascites prescribed propranolol as secondary prophylaxis after first varix bleeding. The average prescription dose of propranolol as secondary prophylaxis was 74 mg/day in patients with large-volume ascites. The mean duration of rebleeding was 22.8 months. Result of analysis showed that low-dose propranolol (40–120 mg/day) compared to inadequate propranolol dose (<40 mg/day) as secondary prophylaxis decreased overall mortality and varix rebleeding in patients with tense ascites. Conclusions: Low-dose propranolol (40–120 mg/day) as secondary prophylaxis for variceal re-bleeding decreased overall mortality and varix rebleeding recurrence in patients with tense ascites.

## 1. Introduction

Variceal bleeding is a major complication of liver cirrhosis [1]. Mortality rate within 6 weeks after the first occurrence of variceal bleeding is approximately 15–20% [2,3]. The incidence of variceal rebleeding within six weeks of the first bleeding is 30% and increases to 60% within one year. The mortality rate of variceal rebleeding is 34% within 12 months [4,5]. Non-selective beta-blockers (NSBBs) can reduce the incidence of variceal rebleeding. Most guidelines recommend the use of beta-blockers as secondary prophylaxis for variceal bleeding [6,7,8].

Recently, physicians are facing controversy on the safety of NSBBs in patients with large-volume ascites. A landmark study by Serste in 2010 first reported that the use of NSBBs increases mortality rate in patients with large-volume ascites [9]. Their follow-up study also showed that the use of NSBBs in patients large-volume ascites was associated with paracentesis-induced circulatory dysfunction [10]. However, other studies reported that NSBBs (at an average dosage of 48.9–80 mg) were relatively effective and safe in patients with large-volume ascites [11,12,13,14]. Thus, most recent reports insisted that small amount of propranolol is known to be safe, if patients didn’t have hypotension, hyponatremia, or impaired renal function [12,15,16,17,18]. However, most previous studies used NSBBs as primary prophylaxis for varix bleeding (never bleeding) in compensated patients or as a mix of primary and secondary prophylaxis. Moreover, sample size of most studies was small.

Therefore, this study aimed to investigate the efficacy and safety of NSBBs as secondary prophylaxis for varix rebleeding in patients with large-volume ascites who require paracentesis.

## 2. Methods

### 2.1. Data Source

This study used data from the Health Insurance Review and Assessment Service (HIRAS). The data are patient claims recorded by the HIRAS, which provides health insurance to 97.2% of the total population of Korea. All insurance claims of hospitals and clinics in Korea are reviewed by the HIRAS. Approximately 46 million claims are filed yearly, including those from >80,000 medical institutions nationwide. These claims are approved and funded by the National Health Insurance Corporation, and data are recorded using encrypted numbers according to the disclosure principle (IRB: HYUH 2017-04-006). All processes were performed in accordance with the relevant guidelines and regulations by both IRB and HIRA. We used open data source, so the informed consent waived.

### 2.2. Study Design 

This is a retrospective cohort study using data from January 2007 to December 2014, as requested by the Health Insurance Review and Assessment Service. Informed consent was waived because of the retrospective nature of the study.

### 2.3. Study Population

A total of 80,071 patients with first varix bleeding as the first decompensated complication enrolled from 2007 to 2014. A total of 27,372 patients with initial variceal hemorrhage were followed up for more than 2 years. Among them, 6826 patients received propranolol for over 30 days after the initial variceal bleeding. This study included, among varix bleeding-naïve patients, 2274 patients with large-volume ascites, which was defined as serum albumin concentration of <3.0 mg/dL and aspirated ascitic fluid of >3 L after paracentesis (Figure 1).

### 2.4. Inclusion Criteria

The inclusion criteria were as follows: (a) patients with variceal bleeding as the first complication and have not been treated for decompensated complications (e.g., varix bleeding, large-volume ascites, hepatic encephalopathy, and hepatorenal syndrome) for the past 2 years; (b) patients treated with the beta-blocker propranolol for more than 30 days.

### 2.5. Exclusion Criteria

The exclusion criteria were as follows: (a) patients who received nadolol or carvedilol, besides propranolol. (b) Patients treated for decompensated complications of liver cirrhosis within 2 years of enrollment. (c) Patients who died within 365 days after variceal bleeding. (d) Patients who were prescribed propranolol for less than 30 days. (e) Patients who were younger than 18 years. (f) Patients diagnosed with malignant tumors within 5 years. (g) Patients who had hepatic encephalopathy or hepatorenal syndrome at the time of variceal bleeding.

### 2.6. Baseline Adjustment of Study Population

Correction of the baseline characteristics in the compared groups was very important in this study. We adopted variceal bleeding as the first complication. Patients who had no decompensated complications within 2 years after diagnosis were included because median survival after first decompensation is approximately 1.6 years. Moreover, patients who died within 1 year after variceal bleeding were excluded to normalize the severity of variceal hemorrhage and disease, and also because the dose of propranolol prescribed for one year after bleeding was calculated using the number of days of medication. Patients who underwent diagnostic paracentesis or those who used diuretics to control ascites were also excluded.

### 2.7. Definition of Inadequate User and Medication Compliance

Patients were included in the inadequate user group if they were prescribed propranolol for more than 30 days and showed inappropriate compliance. However, the average annual prescription dose was <40 mg/day. Medication compliance was calculated as the amount of propranolol prescribed during the first year after variceal bleeding. Only the beta-blocker propranolol was included in the calculation.

### 2.8. Operational Definitions of Decompensated Complications

Decompensated cirrhotic complications were defined as varix bleeding, large-volume ascites, hepatic encephalopathy, and hepatorenal syndrome. Patients with large-volume ascites were defined as those who received paracentesis treatment or were prescribed albumin (227104BIJ) under insurance reimbursement. We gave an insurance code for albumin prescribed to treat volume expansion after large volume paracentesis. Reimbursement condition of albumin is very strict in HIRAS. Albumin use is limited to when blood albumin concentration is <3.0 mg/dL and >3 L after paracentesis. Patients with varix bleeding were defined as those treated with endoscopic sclerotherapy, ligation, or medication (vasopressin, terlipressin, somatostatin, or octreotide). Patients with hepatic encephalopathy and hepatorenal syndrome were defined as those managed using lactulose enema (M0076) and those admitted to a hospital and administered terlipressin and albumin simultaneously as insurance benefits, respectively. Patients were covered by insurance if (a) their serum creatinine level is more than two times higher than the baseline and more than 2.5 mg/dL within 2 weeks, and (b) if their creatinine clearance is reduced by more than 50% over 24 h to <20 mL/min. 

### 2.9. Validation of Operational Definitions

After IRB approval, the medical records of patients from two hospitals were reviewed retrospectively using the operational definitions of varix bleeding, large-volume ascites, and decompensated cirrhosis. A total of 144 patients met the operational definition of variceal bleeding, and 87 patients from two hospital databases met the operative definition of large-volume ascites. The electronic chart of each patient was checked to confirm consistency with the operational definitions (Figure 2).

### 2.10. Primary and Secondary Endpoint 

Primary endpoint was mortality due to the use of the propranolol in patients with simultaneous variceal bleeding and uncontrolled ascites. Secondary endpoint was variceal rebleeding.

### 2.11. Statistical Analysis

The ANOVA test and Chi-square test were used to analyze demographic and biochemical data differences according to sex. Meanwhile, the Kaplan-Meier analysis was used to evaluate the survival rate and frequency of rebleeding with respect to the prescribed dose of propranolol. Data were analyzed with respect to viral and alcohol-induced cirrhosis. 

## 3. Results

### 3.1. Baseline Characteristics

This study included 2274 patients with large-volume ascites who were prescribed propranolol as secondary prophylaxis for more than 30 days (Figure 1). The mean age of the subjects was 52.6 years, and 79.6% of the study population comprised of men. The average prescription dose of propranolol was 74 mg/day (Table 1). The mean follow-up period was 43.7 months, and the mean duration of rebleeding was 22.8 months. Inadequate users (noncompliance group) were defined as those prescribed with an average beta-blocker dose of <40 mg/day, with a mean dose of 31.6 mg/day.

### 3.2. Validation of Operational Definitions

Patient records that met the operational definitions were extracted from two independent hospitals to confirm operational definition and agreement (Figure 2). A total of 144 patient data was extracted from two hospitals as variceal bleeding according to its operational definition, and all those data (100%) were consistent with variceal bleeding caused by liver cirrhosis. Eighty-seven data were extracted from two hospitals as large-volume ascites according to its operational definition. Eighty-five these date (97.7%) were consistent with large-volume ascites. Two inconsistencies were noted. One patient with cirrhosis underwent panperitonitis surgery, but he did not have large-volume ascites. One cirrhotic patient also underwent brain aneurysm surgery.

### 3.3. Effects of Low-Dose Propranolol on Overall Mortality 

The Kaplan-Meier survival curve showed that mortality rate was lower in the low-dose propranolol group (40–120 mg/day) than in the inadequate user group (<40 mg/day) (*p* < 0.001 and *p* = 0.0028, respectively) (Figure 3A). However, this advantage of propranolol was not observed in the high-dose propranolol group (≥120 mg/day). Data were analyzed according to the cause of cirrhosis (viral and non-viral cirrhosis). Propranolol at 40–120 mg/day decreased overall mortality in the viral cirrhosis group (*p* = 0.003, Figure 3B), but and in the non-viral cirrhosis group only propranolol at 40–80 mg/day decreased overall mortality (*p* = 0.006; Figure 3C).

### 3.4. Effects of Low-Dose Propranolol on Rebleeding Rate 

The Kaplan-Meier survival curve showed that propranolol treatment at all dose decreased varix rebleeding rate in patients with tense ascites to a greater extent than inadequate propranolol (<40 mg/day) (Figure 4A). Propranolol decreased varix rebleeding in both patients with viral and non-viral cirrhosis (Figure 4B,C).

## 4. Discussion

This study showed that low-dose propranolol (40–120 mg/day) as secondary prophylaxis decreased overall mortality and recurrence of varix rebleeding. This study used the database of HIRAS, which covers 97.2% of the total population in Korea (*n* = 49,989,620) and is the first large-scale study to identify the effects of propranolol as secondary prophylaxis in patients with large-volume ascites. 

Beta-blockers are a commonly used treatment modality for reducing portal venous pressure and preventing re-bleeding in cirrhotic patients with varix [7,19]. The reason for the use of nonselective beta-blockers in varix is to reduce portal vein pressure by decreasing portal blood flow into the portal vein. Non-selective beta-blockers reduce cardiac output and block the adrenergic dilatory tone of the mesenteric arteriole, leaving only alpha adrenergic-mediated vasoconstriction. It has been used for many years as a pharmacological treatment for prevention of variceal bleeding due to the effect of lowering portal pressure caused by vasoconstriction [20,21].

The prevalence of re-bleeding was low in beta-blocker group in a RCT article, [2] and the meta-analysis showed that beta-blockers were effective in reducing mortality (absolute risk reduction = 7%) [22]. However, in a study published in 2012, the use of beta-blockers in patients with variceal bleeding events showed an increased incidence of death and re-bleeding [9]. The use of beta-blockers in patients with refractory ascites may result in fragile hemodynamic status by a decrease in cardiac output, leading to decreased organ perfusion and increased mortality. Thus, a study suggesting that beta-blockers should not be used in patients with hypotension or organ dysfunction is suggested [23,24]. In subsequent studies, low-dose non-selective beta-blocker is safe in patients with refractory ascites and the effects of beta-blockers with refractory ascites are controversial [25,26]. The purpose of this study was to evaluate the efficacy and safety of beta-blockers in large number patients with variceal bleeding.

In this study, no effect of propranolol was observed according to etiology. In other preliminary studies, no difference in effect was observed when the etiology was calibrated also [2,14,25]. Regardless of etiology of liver cirrhosis, the cause of occurred varix and ascites in patients with liver cirrhosis is the increase of portal venous pressure. The effect of beta-blocker is thought to have no effect on the cause. In addition, propranolol less than 120 mg/day had a beneficial effect in this study, but a beneficial effect was masked at doses greater than 120 mg/day. This tendency is similar to the results of previous studies suggesting that low-dose beta-blockers below 80 mg are safe [26]. Mean dose of NSBB used in Serste’s study [9], which presented a different result from this study, was 113.25 mg, which was higher than the average dose of this article and previous papers. In addition, we think that the severity of the patient was higher than our articles and other papers because patients with Child-Pugh class C were 67.5%, and the mean survival rate was only 8 months. In the previous studies, the relationship between only the mean dose and the mortality of beta-blockers was analyzed. In this paper, we propose clearer cut-off that can be used safely by presenting the amount of beta-blocker usage by intervals. The use of high-dose beta-blockers may further reduce portal venous pressure. Conversely, cardiac reverse may be reduced to increase complications such as acute renal failure or hepatorenal syndrome. It is considered that it is better to pay attention to high dose use of beta-blocker.

The most critical point in using reimbursement claim data is to balance a baseline between comparable groups. Due to the characteristics of the data, there was no lab data, such as albumin or prothrombin time, to evaluate the accurate severity of the patient. In order to supplement this, we attempted to control the patients as homogenously as possible. At first, we included first variceal bleeding as the first decompensation. We reviewed all patients’ reimbursement claim data recorded on the preceding two years. We also excluded the following history of complications within 2 years: tense ascites (paracentesis: C8050, C8051, and Q2470), variceal bleeding (variceal ligation: Q2430-Q2438, Q7631-Q7634), any vasoactive drug (octreotide, vasopressin, terlipressin, or somatostatin), hepatic encephalopathy (lactulose enema: M0076), and hepatorenal syndrome (co-administration of terlipressin and albumin). Second, our definition of tense ascites used was homogenous. Although patients with tense ascites who did not undergo paracentesis or use high-dose diuretics were not included in this study, our definition of tense ascites was uniform. We selected a homogenous population of patients who had large-volume ascites, which we defined as those who were hospitalized and underwent paracentesis with a fluid volume and blood albumin concentration of >3 L and <3.0 mg/dL, respectively. Reimbursement condition of albumin is very strict in HIRAS. All physicians must submit results of serum albumin test as well as paracenthesis code to reimburse albumin. All patients who died within 1 year of variceal hemorrhage were excluded in this study because of the following two reasons. First, we compared mortality and rebleeding rate with medication compliance, whereas medication compliance was calculated on the basis of adherence to medication during the first year after bleeding. For this reason, patients who died within one year were excluded. 

Definition of inadequate group was those who were inadequately prescribed propranolol (<40 mg/day) during the first year, instead of those who did not take propranolol at all. We excluded those who were not prescribed propranolol after variceal bleeding for one year, because they might have had severe co-morbidity (for example, hypotension, chronic obstructive pulmonary disease, diabetes, or other cardiovascular diseases).

To date, several studies on the efficacy and safety of NSBBs have been conducted in patients with large-volume ascites, and the results are conflicting [9,11,12,13,14]. Such results are attributed to the different doses of NSBBs used. Sertes first reported that the use of beta-blockers increases mortality in patients with large-volume ascites [9]. In their study, the mean dose of NSBBs used was 113.25 mg/day. However, in the following three studies, the mean NSBB dosage was <80 mg. On the basis of these data, low-dose (<80 mg) NSBBs are believed to be relatively safe for patients with large-volume ascites [16,26]. Our data supported recent data. We used various cut-off doses of propranolol (40–80 mg/day, 80–120 mg/day, and ≥120 mg/day) to evaluate the benefit of propranolol as secondary prophylaxis in patients with large-volume ascites and found that propranolol at low doses reduced overall mortality and rebleeding rate.

To best our knowledge, the present study had a relatively long follow-up period, with an average of 43.7 months (3.64 years) and included more than 2,000 patients with large-volume ascites. On the contrary, all previous studies had short follow-up periods (<10 months) and a small number of subjects (<150 people).

However, this study had some limitations. First, we could not confirm the medical records of patients because we used data recorded for insurance claims. We also defined large-volume ascites and variceal hemorrhage using their operational definitions. To overcome these problems, we examined the validity of the operational definitions by reviewing medical records from two hospitals. Second, the inadequate user group was defined as the propranolol noncompliance group (inadequate dose of propranolol), instead of the non-propranolol-user group. The mean dose of propranolol in inadequate user group was 31.6 mg because almost all patients took propranolol after variceal bleeding according to the guideline without any contraindication. Patients who did not use NSBBs after variceal bleeding were not included as controls because they were highly likely to have severe cardiovascular and respiratory illnesses or a very poor general condition. As such, the inadequate group was defined as those who received propranolol for more than 30 days after variceal bleeding but were improperly prescribed an average dose of <40 mg. In a previous study, the dose of the appropriate beta-blocker in Korean patients was 154.4 ± 59.4 mg, as measured according to hepatic venous pressure gradient after taking propranolol [27]. Third, we defined propranolol adherence on the basis of the dose of propranolol prescribed at the first year. However, medication compliance at the first year is not representative of compliance during the entire treatment period. In the present study, only medication adherence during the first year was analyzed because if we had calculated the mean dosage of propranolol at the entire study period, it would have affected medication adherence and survival, and subsequently the results of statistical analyses. Fourth, we assumed that effect on short term mortality of other co-morbidity (ex. diabetes, hypertension, and kidney disease) is limited, because life expectancy is very short mean and survival period of decompensated cirrhosis is only two years. But some other information to must be help assess patient severity, for example, comorbidities and comorbidity burden, prior hospitalization/emergency room visits, prior healthcare costs, etc. This would help readers assess whether or not there are differences in patients at baseline and a pre-defined look back period. But we did not address and adjust several important factors. In conclusion, low-dose propranolol (40–120 mg/day) as secondary prophylaxis decreased overall mortality rate in patients with tense ascites. This has potential applications for physicians in clinical practice. Secondary prophylaxis using low-dose propranolol (40–120 mg/day) after variceal bleeding can be safe even if patients have large-volume ascites.

## Figures and Tables

**Figure 1 jcm-08-00573-f001:**
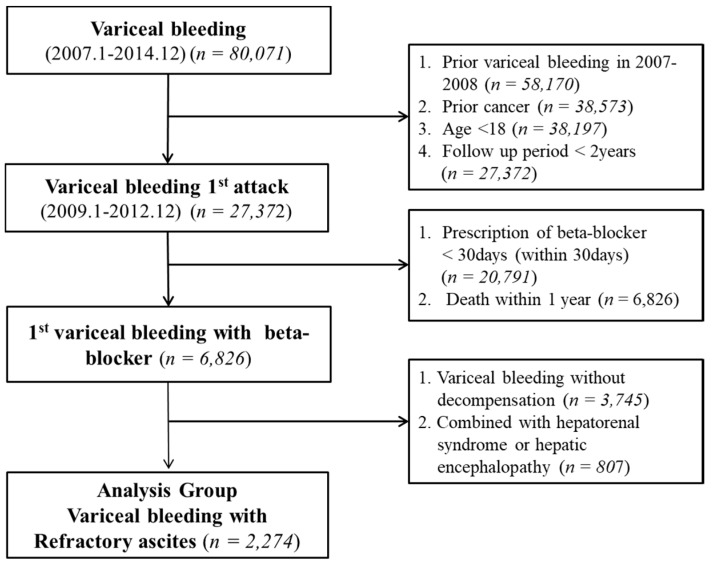
Flowchart.

**Figure 2 jcm-08-00573-f002:**
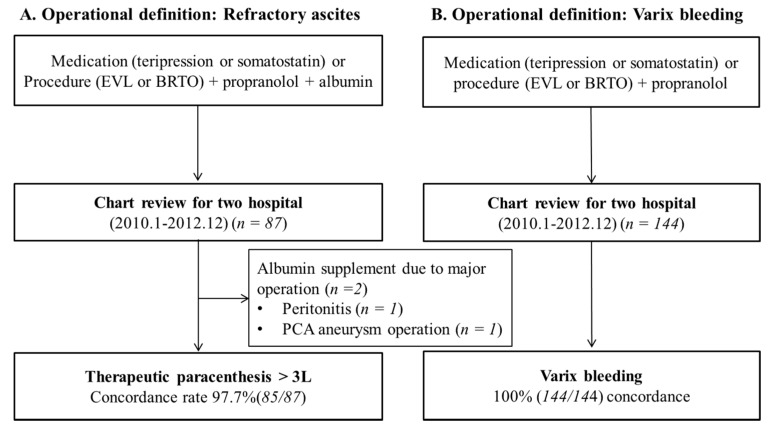
Validation of operational definition. (**A**) Variceal bleeding with large-volume ascites, (**B**) Variceal bleeding.

**Figure 3 jcm-08-00573-f003:**
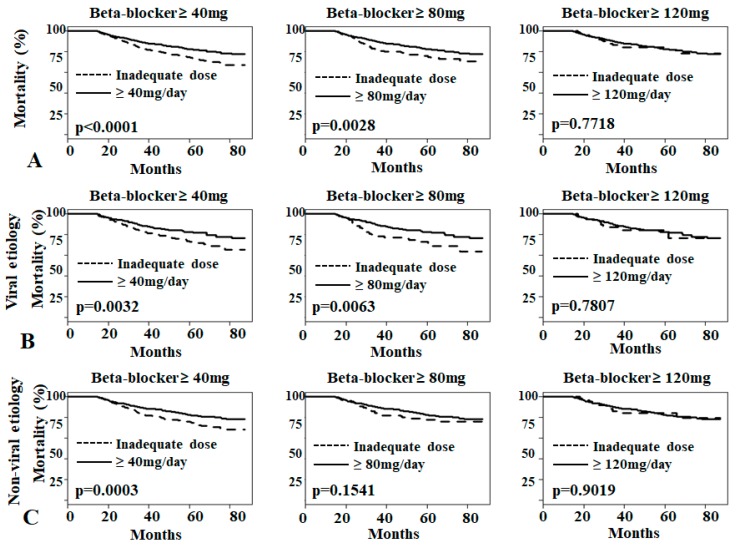
Mortality due to various doses of non-selective beta-blockers in patients with large-volume ascites. (**A**) Overall mortality rate due to various dosages of non-selective beta-blockers (NSBB). (**B**) Overall mortality rate due to various dosages of NSBB in patients with viral cirrhosis. (**C**) Overall mortality rate due to various dosages of NSBB in patients with non-viral cirrhosis.

**Figure 4 jcm-08-00573-f004:**
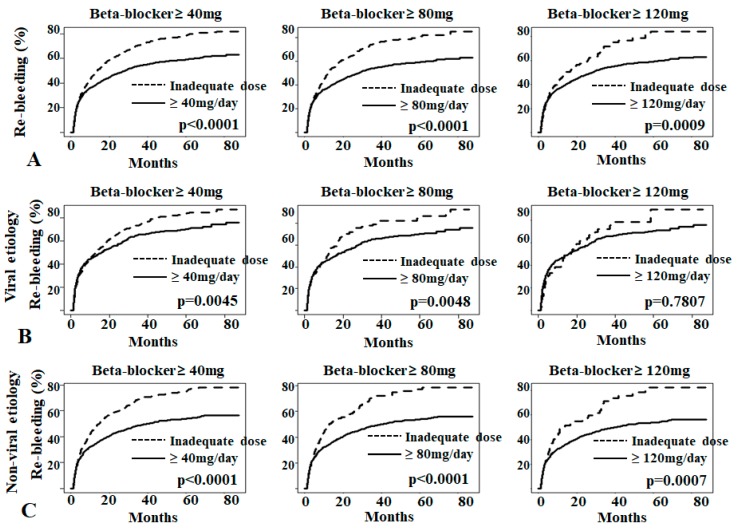
Rebleeding in patients with variceal bleeding and large-volume ascites. (**A**) Varix rebleeding rate due to various dosages of NSBB. (**B**) Varix rebleeding rate due to various dosages of NSBB in patients with viral cirrhosis. (**C**) Varix rebleeding rate due to various dosages of NSBB in patients with non-viral cirrhosis.

**Table 1 jcm-08-00573-t001:** Basic characteristics of patients.

Variables	All (*n* = 2274)	Inadequate (*n* = 1388)	Propranolol 40–80 mg (*n* = 608)	Propranolol 80–120 mg (*n* = 177)	Propranolol >120 mg (*n* = 101)	* *p*-Value
Age	52.60 ± 11.74	52.94 ± 12.07	52.59 ± 11.04	51.85 ± 11.78	49.33 ± 10.71	<0.0001
Follow up period (month)	43.74 ± 15.82	44.26 ± 15.49	43.15 ± 16.39	42.04 ± 16.35	43.12 ± 15.81	0.2091
Rebleeding days (month)	22.84 ± 21.02	25.35 ± 21.88	19.40 ± 19.17	17.36 ± 8.76	18.75 ± 18.04	<0.0001
Dosage of β-blocker (mg)	74 ± 45.6	31.6 ± 41.6	76.4 ± 32.4	109.6 ± 29.6	169.6 ± 42.4	<0.0001
Men (%)	1810 (79.60)	1094 (78.82)	491 (80.76)	142 (80.23)	83 (82.18)	0.6872
Etiology (%)						
HBV	673 (29.60)	373 (26.87)	201 (33.06)	66 (37.29)	33 (32.67)	0.0030
HCV	186 (8.18)	100 (7.20)	57 (9.38)	23 (12.99)	6 (5.94)	0.0286
Non-viral	1415 (62.22)	915 (65.93)	350 (57.56)	88 (49.72)	62 (61.39)	0.0002

* *p*-value < 0.05 is statistically significant (ANOVA test); HBV, hepatitis B virus; HCV, hepatitis C virus.

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
