# Peer review of "Low-Dose Propranolol as Secondary Prophylaxis for Varix Bleeding Decreases Mortality and Rebleeding Rate in Patients with Tense Ascites"

_jcm, 2019, doi:10.3390/jcm8050573_

Reviewer 1 Report

Overall - the study is clearly presented, detailed results figures are greatly appreciated, and I believe the article addresses an interesting clinical question.

The primary concern is related to confounding related to choice of dose and choice of treatment. Primarily, it does not appear that the authors have controlled for any measure of disease severity. Hypothetically, disease severity may influence the use of propanolol and the dose received? If so, then having some control for this is needed. This could be why the results are reversed between the univariate and multivariable analysis (*please also note that the correct terminology is "multivariable"). Further, as the results are flipped, it is somewhat misleading to have so many figures focused on the unadjusted results. As such, I would highly recommend presenting some adjusted survival curves in addition to equally weight the adjusted results. This could easily be done from the Cox PH model. There is also some inconsistent references to the "inadequate" group as the "control" grou and this should be kept consistent. It also appears you have presented the etiology results in table 1 with row percentages, please change to column percentages.

Author Response

Overall - the study is clearly presented, detailed results figures are greatly appreciated, and I believe the article addresses an interesting clinical question.

The primary concern is related to confounding related to choice of dose and choice of treatment. Primarily, it does not appear that the authors have controlled for any measure of disease severity. Hypothetically, disease severity may influence the use of propranolol and the dose received? If so, then having some control for this is needed. This could be why the results are reversed between the univariate and multivariable analysis (*please also note that the correct terminology is "multivariable").

-> First, thank you for pointing out. As you mentioned, controlling disease severity is a critical problem. This study was covered about 97.2% of Koreans, but the database characteristics of this epidemiologic study, there was no lab data, such as albumin or prothrombin time, to evaluate the accurate severity of the patient. So, as you mentioned, it seems that there is not enough tool to assess disease severity.

Therefore, we used the following method to overcome the above problem.

First, patients in most of the previous studies used beta-blockers for primary and secondary prophylaxis for variceal bleeding. However, this study targeted patients who used beta-blockers for secondary prophylaxis only. Second, patients with first decompensated event of variceal bleeding were included study only, and patients who had other decompensated event within 2 years or who died within 1 year of enrollment were excluded. Third, beta-blocker has been used to prevent variceal bleeding for patients who were reported variceal bleeding or observed varix greater than F2 grade. The reason of not using the beta-blocker is mostly because of the cardiopulmonary disease which is contraindication of beta-blocker. Thus, this study targeted only patients using beta-blockers for at least 1 week and attempted to compensate for these areas. In addition, the control group was defined as a group that did not use beta-blockers and analysis was performed in advance. The short-term mortality rate was high in the group without beta-blockers. Based on these results, it was judged that there was contraindication of Propranolol such as Comorbidity when Propranolol was not used. Therefore, patients not using propranolol were not defined as control group but inadequate dose was defined as control group.

Using these methods, we tried to adjust the patients as homogenously and control the enroll criteria as strictly as possible. As a result, among the 80071 patients who had variceal bleeding from 2007 to 2014, the final enrolled population was 2,224, with only 2.8% enrolled. We tried to homogenously control the patient population as much as possible. Related parts were mentioned from page 9, line 211 to page 9, line 228 and page 10, line 248 to page 10, line 252..

… At first, we included first variceal bleeding as the first decompensation. We reviewed all patients' reimbursement claim data recorded on the preceding two years. We also excluded the following history of complications within 2 years: tense ascites (paracentesis: C8050, C8051, and Q2470), variceal bleeding (variceal ligation: Q2430-Q2438, Q7631-Q7634), any vasoactive drug (octreotide, vasopressin, terlipressin, or somatostatin), hepatic encephalopathy (lactulose enema: M0076), and hepatorenal syndrome (co-administration of terlipressin and albumin). Second, our definition of tense ascites used was homogenous. Although patients with tense ascites who did not undergo paracentesis or use high-dose diuretics were not included this study, our definition of tense ascites was uniform. We selected a homogenous population of patients who had large-volume ascites, which we defined as those who were hospitalized and underwent paracentesis with a fluid volume and blood albumin concentration of >3 L and<3.0 mg/dL, respectively. Reimbursement condition of albumin is very strict in HIRAS. All physicians must submit results of serum albumin test as well as paracentesis code to reimburse albumin. All patients who died within 1 year of variceal hemorrhage were excluded in this study because of the following two reasons. First, we compared mortality and rebleeding rate with medication compliance, whereas medication compliance was calculated on the basis of adherence to medication during the first year after bleeding. For this reason, patients who died within one year were excluded.

However, this study had some limitations. First, we could not confirm the medical records of patients because we used data recorded for insurance claims. We also defined large-volume ascites and variceal hemorrhage using their operational definitions. To overcome these problems, we examined the validity of the operational definitions by reviewing medical records from two hospitals.

 Further, as the results are flipped, it is somewhat misleading to have so many figures focused on the unadjusted results. As such, I would highly recommend presenting some adjusted survival curves in addition to equally weight the adjusted results. This could easily be done from the Cox PH model.

-> As you pointed out, the results of the multivariate analysis were reversed compared to results in univariate analysis that the use of low-dose propranolol increases mortality and re-bleeding. After checking you review, we re-evaluated the overall methodology of multivariate analysis with statistical experts. It is somewhat risk to perform a multivariate analysis with only identifiable variables, and some of the variables could not guarantee a proportional hazard. The interpretation of the results was discussed again with the statistical experts and we decided to delete and revised contents about multivariate analysis because the interpretation of the results was considered necessary. Since there are not enough to control variables and some of the variables that can be investigated are not suitable for the Cox proportional hazard model, it has been determined that presenting descriptive results as whole investigations could deliver error-free results. Although we did not conduct multivariate analysis, this study is a comprehensive survey of about 97.2% of the nation's population. It is also meaningful to present a descriptive result by strictly managing the enroll criteria.

 There is also some inconsistent references to the "inadequate" group as the "control" group and this should be kept consistent. It also appears you have presented the etiology results in table 1 with row percentages, please change to column percentages.

-> In order to make the terms uniform, they were unified into inadequate group and the points in Table 1 were revised. Related changes are marked in red.

Reviewer 2 Report

1.      The methodology is questionable. Collinearity diagnosis/multicollinearity could be present in multivariate regression to lead to totally reverse consequence compared to those by univariate regression. This could  result in unrealistic and untenable interpretations (Kristina P. Vatcheva et al., 2016). Please recheck the overall methodology, adjustment of multivariate analysis and select the more appropriate predictor variables (Kristina P. Vatcheva et al., 2016; Priya Ranganathan et al., 2017).

2.      The discussion is too short, shallow and not extensive enough for SCI like JCM. Etiology, possible mechanisms and effects of dose of the use of nonselective beta-blockers in patients with different types of ascites should be included to analyze and evaluate the overall statistical data in the study.

 References:

1.      Kristina P. Vatcheva, MinJae Lee, Joseph B. McCormick, and Mohammad H. Rahbar,Epidemiology (Sunnyvale). 2016 Apr; 6(2): 227. Multicollinearity in Regression Analyses Conducted in Epidemiologic Studies

2.      Priya Ranganathan, C. S. Pramesh, and Rakesh Aggarwal. Perspect Clin Res. 2017 Jul-Sep; 8(3): 148–151. Common pitfalls in statistical analysis: Logistic regression

Author Response

1.      The methodology is questionable. Collinearity diagnosis/multicollinearity could be present in multivariate regression to lead to totally reverse consequence compared to those by univariate regression. This could result in unrealistic and untenable interpretations (Kristina P. Vatcheva et al., 2016). Please recheck the overall methodology, adjustment of multivariate analysis and select the more appropriate predictor variables (Kristina P. Vatcheva et al., 2016; Priya Ranganathan et al., 2017).

-> First, thank you for pointing out. After checking you review, we re-evaluated the overall methodology of multivariate analysis with statistical experts.  As you pointed out, the results of the multivariate analysis were reversed compared to results in univariate analysis that the use of low-dose propranolol increases mortality and re-bleeding. It is somewhat risk to perform a multivariate analysis with only few identifiable variables, and some of the variables could not guarantee a proportional hazard. The interpretation of the results was discussed again with the statistical experts and we decided to delete and revised contents about multivariate analysis because the interpretation of the results was considered necessary and maintaining of multivariate analysis is risky.

Since there are not enough to control variables and some of the variables that can be investigated are not suitable for the Cox PH model, it has been determined that presenting descriptive results as whole investigations could deliver error-free results. Although we did not conduct multivariate analysis, this study is a comprehensive survey of about 97.2% of the nation's population. It is also meaningful to present a descriptive result by strictly managing the enroll criteria. Although our data was not sufficient to do multivariate analysis because there was no lab data, such as albumin or prothrombin time, we used the following method to overcome several confounding factors and adjust baseline characteristics.

First, most of the existing studies have used beta-blockers for primary and secondary prophylaxis for variceal bleeding. However, this study targeted patients who used beta-blockers for secondary prophylaxis only. Second, patients with first decompensated event of variceal bleeding were included study only, and patients who had other decompensated event within 2 years or who died within 1 year of enrollment were excluded. Third, beta-blocker has been used to prevent variceal bleeding for patients who were reported variceal bleeding or observed varix greater than F2 grade. The reason of not using the beta-blocker is mostly because of the cardiopulmonary disease which is contraindication of beta-blocker. Thus, this study targeted only patients using beta-blockers for at least 1 week and attempted to compensate for these areas. In addition, the control group was defined as a group that did not use beta-blockers and analysis was performed in advance. The short-term mortality rate was high in the group without beta-blockers. Based on these results, it was judged that there was contraindication of Propranolol such as Comorbidity when Propranolol was not used. Therefore, patients not using propranolol were not defined as control group but inadequate dose was defined as control group.

Related parts were mentioned from page 9, line 211 to page 9, line 228 and page 10, line 248 to page 10, line 252.

… At first, we included first variceal bleeding as the first decompensation. We reviewed all patients' reimbursement claim data recorded on the preceding two years. We also excluded the following history of complications within 2 years: tense ascites (paracentesis: C8050, C8051, and Q2470), variceal bleeding (variceal ligation: Q2430-Q2438, Q7631-Q7634), any vasoactive drug (octreotide, vasopressin, terlipressin, or somatostatin), hepatic encephalopathy (lactulose enema: M0076), and hepatorenal syndrome (co-administration of terlipressin and albumin). Second, our definition of tense ascites used was homogenous. Although patients with tense ascites who did not undergo paracentesis or use high-dose diuretics were not included this study, our definition of tense ascites was uniform. We selected a homogenous population of patients who had large-volume ascites, which we defined as those who were hospitalized and underwent paracentesis with a fluid volume and blood albumin concentration of >3 L and<3.0 mg/dL, respectively. Reimbursement condition of albumin is very strict in HIRAS. All physicians must submit results of serum albumin test as well as paracentesis code to reimburse albumin. All patients who died within 1 year of variceal hemorrhage were excluded in this study because of the following two reasons. First, we compared mortality and rebleeding rate with medication compliance, whereas medication compliance was calculated on the basis of adherence to medication during the first year after bleeding. For this reason, patients who died within one year were excluded.

However, this study had some limitations. First, we could not confirm the medical records of patients because we used data recorded for insurance claims. We also defined large-volume ascites and variceal hemorrhage using their operational definitions. To overcome these problems, we examined the validity of the operational definitions by reviewing medical records from two hospitals.

 2.      The discussion is too short, shallow and not extensive enough for SCI like JCM. Etiology, possible mechanisms and effects of dose of the use of nonselective beta-blockers in patients with different types of ascites should be included to analyze and evaluate the overall statistical data in the study.

-> Beta-blockers are a commonly used treatment modality for reducing portal venous pressure and preventing re-bleeding in cirrhotic patients with varix. First, no effect of propranolol was observed according to etiology in this study.1-3. There were no differences in effect when the etiology was adjusted also in other previous studies. Regardless of etiology of liver cirrhosis, the cause of occurred varix and ascites in patients with liver cirrhosis is the increase of portal venous pressure. The effect of beta-blocker is thought to have no effect on the cause.

Second, propranolol less than 120 mg/day had a beneficial effect in this study, but a beneficial effect was masked at doses greater than 120 mg/day. This tendency is similar to the results of previous studies suggesting that low-dose beta-blockers about 80 mg are safe.4 Mean dose of NSBB used in Serste's study5, which presented a different result from this study, was 113.25 mg, which was higher than the average dose of this article and previous papers. In addition, we think that the severity of the patient was higher than our articles and other papers because patients with Child-Pugh class C were 67.5%, and the mean survival rate was only 8 months. In the previous studies, the relationship between only the mean dose and the mortality of beta-blockers was analyzed. In this paper, we propose clearer cut-off that can be used safely by presenting the amount of beta blocker usage by intervals. The use of high-dose beta-blockers may further reduce portal venous pressure. Conversely, cardiac reverse may be reduced to increase complications such as acute renal failure or hepatorenal syndrome. It is considered that it is better to pay attention to high dose use of beta-blocker.

1. Poynard T, Cales P, Pasta L, Ideo G, Pascal JP, Pagliaro L, Lebrec D: Beta-adrenergic-antagonist drugs in the prevention of gastrointestinal bleeding in patients with cirrhosis and esophageal varices. An analysis of data and prognostic factors in 589 patients from four randomized clinical trials. Franco-Italian Multicenter Study Group. The New England journal of medicine 1991, 324(22):1532-1538.

2. Onali S, Kalafateli M, Majumdar A, Westbrook R, O'Beirne J, Leandro G, Patch D, Tsochatzis EA: Non-selective beta-blockers are not associated with increased mortality in cirrhotic patients with ascites. Liver international : official journal of the International Association for the Study of the Liver 2017, 37(9):1334-1344.

3. Brito-Azevedo A: Carvedilol and survival in cirrhosis with ascites: A cognitive bias? Journal of hepatology 2017, 67(2):425-426.)

4. Reiberger T, Mandorfer M: Beta adrenergic blockade and decompensated cirrhosis. Journal of hepatology 2017, 66(4):849-859.

5. Serste T, Melot C, Francoz C, Durand F, Rautou PE, Valla D, Moreau R, Lebrec D: Deleterious effects of beta-blockers on survival in patients with cirrhosis and refractory ascites. Hepatology 2010, 52(3):1017-1022.

Related parts were added in red from page 8, line 181 to page 9, line 207..

… Beta-blockers are a commonly used treatment modality for reducing portal venous pressure and preventing re-bleeding in cirrhotic patients with varix. The prevalence of re-bleeding was low in beta-blocker group in a RCT article, and the meta-analysis showed that beta blockers were effective in reducing mortality (absolute risk reduction = 7%). However, in a study published in 2012, the use of beta-blockers in patients with variceal bleeding events showed an increased incidence of death and re-bleeding. There is also a study that says that the cardiac reverse can be reduced and the mortality rate can be increased in patients with refractory ascites. In subsequent studies, the effects of beta-blockers and their safety on outcomes in patients with refractory ascites are controversial. The purpose of this study was to evaluate the efficacy and safety of beta-blockers in large number patients with variceal bleeding.

In this study, no effect of propranolol was observed according to etiology. In other preliminary studies, no difference in effect was observed when the etiology was calibrated also. Regardless of etiology of liver cirrhosis, the cause of occurred varix and ascites in patients with liver cirrhosis is the increase of portal venous pressure. The effect of beta-blocker is thought to have no effect on the cause. In addition, propranolol less than 120 mg/day had a beneficial effect in this study, but a beneficial effect was masked at doses greater than 120 mg/day. This tendency is similar to the results of previous studies suggesting that low-dose beta-blockers below 80 mg are safe. Mean dose of NSBB used in Serste's study, which presented a different result from this study, was 113.25 mg, which was higher than the average dose of this article and previous papers. In addition, we think that the severity of the patient was higher than our articles and other papers because patients with Child-Pugh class C were 67.5%, and the mean survival rate was only 8 months. In the previous studies, the relationship between only the mean dose and the mortality of beta-blockers was analyzed. In this paper, we propose clearer cut-off that can be used safely by presenting the amount of beta blocker usage by intervals. The use of high-dose beta-blockers may further reduce portal venous pressure. Conversely, cardiac reverse may be reduced to increase complications such as acute renal failure or hepatorenal syndrome. It is considered that it is better to pay attention to high dose use of beta-blocker.

Round  2

Reviewer 1 Report

While I appreciate the attempts to adress my prior concerns, I'm not too sure the response is adequate. While you don't have any lab data to use, you could still provide some other information to help assess patient severity, for example, comorbidities and comorbidity burden, prior hospitalization/ER visits, prior healthcare costs, etc. This would help me assess whether or not there are differences in patients at baseline and a pre-defined look back period (e.g. 6 or 12 months) leading up to the treatment decision of inadequate or adequate. This is pivotal if we are to rely on unadjusted results. 

Author Response

Thank you for your good comment. We tried to adjusted the baseline of disease severity via excluding patients who died within one year and patients with a decompensated event within two years. We used very strict inclusion criteria. We enrolled first decompensation patients presenting with varix bleeding and massive ascites. It is very well known that mean survival period of decompensated cirrhosis is only two years. Because life expectancy and median survival rate is very short, we assumed that effect on short term mortality of other co-morbidity (ex. diabetes, hypertension, and kidney disease) is limited relatively. But our team and statistician fully agree with your great point. If we did adjust confounding factors (for example, comorbidities and comorbidity burden, prior hospitalization/ER visits, prior healthcare costs, etc) as your guidance, it must be help readers assess whether or not there are differences in patients at baseline and improve quality of study. So our team and statistician asked HIRA to re-assess data, but it was not possible to re-connect to the database and extract the data because of HIRA’s policy. So we couldn’t do multivariate analysis to adjust important confounding factors (comorbidities and comorbidity burden, prior hospitalization/ER visits, prior healthcare costs, etc) according to your suggestion. So we added your comments and suggestion as our study limitation in discussion part. We deeply regret that we could not correct your point. Related parts were mentioned from page 10, line 275 to page 10, line 281.

… Fourth, we assumed that effect on short term mortality of other co-morbidity (ex. diabetes, hypertension, and kidney disease) is limited, because life expectancy is very short mean and survival period of decompensated cirrhosis is only two years. But some other information to must be help assess patient severity, for example, comorbidities and comorbidity burden, prior hospitalization/emergency room visits, prior healthcare costs, etc. This would help readers assess whether or not there are differences in patients at baseline and a pre-defined look back period. But we did not address and adjust several important factors.

Reviewer 2 Report

1. The methodology is questionable. Collinearity diagnosis/multicollinearity could be present in multivariate regression to lead to totally reverse consequence compared to those by univariate regression. This could result in unrealistic and untenable interpretations (Kristina P. Vatcheva et al., 2016). Please recheck the overall methodology, adjustment of multivariate analysis and select the more appropriate predictor variables (Kristina P. Vatcheva et al., 2016; Priya Ranganathan et al., 2017).

Comments: The totally reverse consequences were obtained by authors re-evaluated the overall methodology of multivariate analysis with statistical experts. The revised data looked more reasonable but the paper and new results are recommended to be reviewed by another researcher with expertise in epidemiology regarding human medicine.

2. The discussion is too short, shallow and not extensive enough for SCI like JCM.

Etiology, possible mechanisms and effects of dose of the use of non-selective beta blockers in patients with different types of ascites should be included to analyze and evaluate the overall statistical data in the study.

Comments: Dose effects of propranolol have been found in Orthopedics and pharmacological areas. Please discuss the possible mechanisms and evaluate/conclude the use of benefits/disadvantages of selective beta blockers/ non-selective beta blockers in cirrhotic patients and explain those like “why cardiac reverse may be reduced to increase complications such as acute renal failure or hepatorenal syndrome.”

Author Response

Comments: The totally reverse consequences were obtained by authors re-evaluated the overall methodology of multivariate analysis with statistical experts. The revised data looked more reasonable but the paper and new results are recommended to be reviewed by another researcher with expertise in epidemiology regarding human medicine.

☞ We discussed the revised result and methodology with relevant experts. If we could adjust several important confounding factors in this study such as comorbidities and comorbidity burden, prior hospitalization, prior healthcare costs, etc., it must be help us and readers assess whether or not there are differences in patients at baseline and a pre-defined look back period. But statistician recommend not to include the incomplete multivariate analysis, because several variables were not guaranteed proportional hazard assumption. We also discussed with epidemiologist. Epidemiologist pointed out that although we couldn’t do multivariate analysis, descriptive analysis itself could have a meaning because this data cover 97.2% of the total population of Korea. 

Please discuss the possible mechanisms and evaluate/conclude the use of benefits/disadvantages of selective beta blockers/ non-selective beta blockers in cirrhotic patients and explain those like “why cardiac reverse may be reduced to increase complications such as acute renal failure or hepatorenal syndrome.”

☞ The reason for the use of nonselective beta blockers in varix is to reduce portal vein pressure by decreasing portal blood flow into the portal vein. Non-selective beta-blockers reduce cardiac output and block the adrenergic dilatory tone of the mesenteric arteriole, leaving only alpha adrenergic mediated vasoconstriction. It has been used for many years as a pharmacological treatment for prevention of variceal bleeding due to the effect of lowering portal pressure caused by vasoconstriction.1,2

However, in a 2012 report, an increased mortality rate was reported for non-selective beta blockers in patients with refractory ascites. The use of beta blockers in patients with refractory ascites may result in fragile hemodynamic status by a decrease in cardiac output, leading to decreased organ perfusion and increased mortality. Thus, a study suggesting that beta-blockers should not be used in patients with hypotension or organ dysfunction is suggested.3,4

Recent studies have shown that low-dose non-selective beta-blocker is safe in patients with refractory ascites.5,6 The effects of beta-blockers and their safety on outcomes in patients with refractory ascites are controversial. The purpose of this study was to evaluate the efficacy and safety of beta-blockers in large number patients with variceal bleeding.

1. Feu F, García-Pagán JC, Bosch J, et al. haemorrhage in patients with cirrhosis. Lancet 1995;346:1056–1059.

2. Abraldes JG, Tarantino I, Turnes J, et al. Hemodynamic response to pharmacological treatment of portal hypertension and longterm prognosis of cirrhosis. Hepatology 2003;37:902–908.

3. Garcia-Tsao G. Beta blockers in cirrhosis: J Hepatol. 2016 Mar;64(3):532-4. doi: 10.1016/j.jhep.2015.12.012. Epub 2015 Dec 24.

4. Ge PS, Runyon BA. The changing role of beta-blocker therapy in patients with cirrhosis. J Hepatol. 2014 Mar;60(3):643-53. doi: 10.1016/j.jhep.2013.09.016. Epub 2013 Sep 26.

5. Brito-Azevedo A: Carvedilol and survival in cirrhosis with ascites: A cognitive bias? Journal of hepatology 2017, 67(2):425-426.

6. Reiberger T, Mandorfer M: Beta adrenergic blockade and decompensated cirrhosis. Journal of hepatology 2017, 66(4):849-859.

Related parts were mentioned from page 8, line 182 to page 9, line 198.

… Beta-blockers are a commonly used treatment modality for reducing portal venous pressure and preventing re-bleeding in cirrhotic patients with varix. The reason for the use of non-selective beta blockers in varix is to reduce portal vein pressure by decreasing portal blood flow into the portal vein. Non-selective beta-blockers reduce cardiac output and block the adrenergic dilatory tone of the mesenteric arteriole, leaving only alpha adrenergic mediated vasoconstriction. It has been used for many years as a pharmacological treatment for prevention of variceal bleeding due to the effect of lowering portal pressure caused by vasoconstriction.